# Warburg Effect, Glutamine, Succinate, Alanine, When Oxygen Matters

**DOI:** 10.3390/biology10101000

**Published:** 2021-10-04

**Authors:** Frédéric Bouillaud, Noureddine Hammad, Laurent Schwartz

**Affiliations:** 1Institut Cochin, INSERM, CNRS, Université de Paris, F-75014 Paris, France; noureddine.hammad@inserm.fr; 2Assistance Publique des Hôpitaux de Paris, Avenue Victoria, 75003 Paris, France; dr.laurentschwartz@gmail.com

**Keywords:** mitochondria, glycolysis, lactic fermentation, ATP, energy metabolism, inflammation, cancer

## Abstract

**Simple Summary:**

The “Warburg effect” refers to the situation wherein cellular energetics (ATP formation) use “aerobic glycolysis” (i.e., glucose use with the release of lactate (2 ATP per glucose)) even if oxygen present would authorize full oxidation with a much higher yield (34 ATP per glucose). The present article reviews possible reasons to explain this metabolic bias.

**Abstract:**

Cellular bioenergetics requires an intense ATP turnover that is increased further by hypermetabolic states caused by cancer growth or inflammation. Both are associated with metabolic alterations and, notably, enhancement of the Warburg effect (also known as aerobic glycolysis) of poor efficiency with regard to glucose consumption when compared to mitochondrial respiration. Therefore, beside this efficiency issue, other properties of these two pathways should be considered to explain this paradox: (1) biosynthesis, for this only indirect effect should be considered, since lactate release competes with biosynthetic pathways in the use of glucose; (2) ATP production, although inefficient, glycolysis shows other advantages when compared to mitochondrial respiration and lactate release may therefore reflect that the glycolytic flux is higher than required to feed mitochondria with pyruvate and glycolytic NADH; (3) Oxygen supply becomes critical under hypermetabolic conditions, and the ATP/O_2_ ratio quantifies the efficiency of oxygen use to regenerate ATP, although aerobic metabolism remains intense the participation of anaerobic metabolisms (lactic fermentation or succinate generation) could greatly increase ATP/O_2_ ratio; (4) time and space constraints would explain that anaerobic metabolism is required while the general metabolism appears oxidative; and (5) active repression of respiration by glycolytic intermediates, which could ensure optimization of glucose and oxygen use.

## 1. Introduction

There are challenges whose issue (survival or death) depends on adaptive response in the short term, which is too short for reprogramming of gene expression. One of these challenges is the lack of metabolic energy. Cellular bioenergetics extracts energy from the environment to phosphorylate ADP into ATP known as the “energetic currency of the cell” (abbreviations are explained in Appendix A). The cellular content in ATP would cover at most a few minutes of energy requirements for cell survival. Therefore, regeneration of ATP with adaptation of cellular bioenergetics to environmental conditions is an absolute requirement in the short term. For mammalian cells, a simple description would state that mitochondrial respiration and lactic fermentation regenerate ATP to feed cellular bioenergetics.

The yield of respiration and of lactic fermentation could be compared based on the use of one glucose molecule. Lactic fermentation regenerates two ATPs per glucose and releases two molecules of lactic acid. Respiration needs, in addition, six molecules of oxygen (O_2_), and if the yield is 100% it regenerates thirty-four ATP per glucose with the release of six CO_2_ and twelve H_2_O. While lactic fermentation is bound to the use of glucose, the oxidative metabolism may oxidize a large number of organic molecules; and therefore, when no substrates is found in the environment the cell becomes the fuel for the cell (autophagy).

At the beginning of the twentieth-century, Otto Warburg coined the paradox that mammalian cells, and particularly cancer cells, in the presence of oxygen continue to use inefficient lactic acid fermentation. The term “Warburg effect“ or “aerobic glycolysis” is used to refer to this phenomenon [1]. An abundant literature highlights this characteristic of immune cells as well as of cancerous cells. Therefore, driving forces are thought to drive this “metabolic bias”. This paper presents an overview of different possible explanations for this phenomenon.

## 2. Biosynthesis

This proposal gives a “positive value” that balances the disadvantage of recruitment of a low efficiency pathway in terms of cellular bioenergetics and, moreover, it fits with the increased demand in biosynthetic intermediates required by dividing cancer cells. However, it hardly resists a closer look (Appendix A); the final product lactic acid characterizes aerobic glycolysis and there is no change in carbon content of the substrate glucose (C_6_) when compared to the final product (two lactic acids = 2 × C_3_). In other words, for a given cell, the diversion of glycolytic intermediates to biosynthesis would decrease lactic acid release. Therefore, they are in direct competition for the use of glucose. Moreover, for a net ATP synthesis, glycolysis has to go up to its end (i.e., formation of pyruvate). The fate of this pyruvate would be either the formation of lactic acid or introduction in other metabolic pathways (such as the TCA cycle) to generate other biosynthetic intermediates, such as citrate for the formation of lipids and/or to increase ATP production. This role of mitochondrial metabolism has already been highlighted [2]. Then, an explanation for aerobic glycolysis would be that the diversion of glycolytic intermediates to biosynthetic pathways requires an increase in their concentrations, including that of pyruvate, which would promote the activity of lactate dehydrogenase (LDH) to generate lactate and its export out of the cell. Then, within a single cell, lactic acid release represents a price to pay more than a factor promoting biosynthesis. If fluxes are considered, the ATP requirement is likely to generate a lactate efflux much larger than the flux of biosynthetic pathways.

## 3. ATP Production

Respiration is much more efficient and flexible with regard to substrates. However, it has two potential weaknesses. The first is the need of oxygen, whose supply (see below) or presence for oxygen sensitive cellular sites/activities might be a problem, and the second is the complexity of the machinery involved.

Mitochondrial oxidative phosphorylation (oxphos) requires cooperation of five membranous enzymatic complexes (*complexes I*–*V*) approaching a million Dalton each. Moreover, the exchange of ADP against ATP (500 Daltons) across mitochondrial membranes and their diffusion to/from the site of consumption is needed. When proximity between ATP production and consumption is required, the couple of a glycolytic ATP generating step and its substrate (a small fast diffusing molecule) would improve mobility or performance at the expense of yield [3]. Glycolysis starts with activation of sugar by phosphorylation with consumption of two ATP per glucose. If this activation takes place with mitochondrial ATP, net ATP release starts from the first glycolytic ATP by the phosphoglycerate kinase (PK) reaction. Notably, hexokinase, the first ATP using enzyme of glycolysis, was found to be associated with mitochondria [4]. Then, rather than a lactic fermentation compensating for deficient mitochondria, mitochondrial oxphos would actually assist the localized glycolytic ATP production by providing the ATP required to activate glucose. This localized glycolytic ATP generation may then release NADH and pyruvate in amounts that exceed mitochondrial ability/need to oxidize them, hence causing lactate release, even if oxygen supply is sufficient [5].

If transient surges in ATP production are considered the energy cost for building and maintenance of “a mitochondrial reserve” might not be worth the improvement in yield [6,7], and particularly in a complex organism, since lactate would constitute a highly valuable oxidative substrate for other cells/organs [8].

The complexity of mitochondrial bioenergetics makes it potentially sensitive to a large number of adverse conditions. On one side the number of possible targets (individual proteins) in the mitochondrial respiratory chain is large, and on the other side the convergence of all significant mitochondrial metabolic oxidation pathways to the reduction of quinone in the hydrophobic environment of mitochondrial inner membrane makes oxphos a target for a large number of hydrophobic/amphiphilic “membrane troublemakers”. As a consequence, mitochondrial toxicity is a property shared by a large number of small/middle size molecules (drugs) [9,10,11,12]. Cationic amphiphilic drugs are known to cause mitochondrial dysfunction in the liver [12]. This is explained by the mitochondrial membrane potential expected to increase by orders of magnitude the concentration of a permeant cation, hence increasing greatly the exposure of intramitochondrial enzymes to otherwise weak inhibitors. Moreover, a number of pathogens impact on mitochondrial bioenergetics [13,14,15]. Then, aerobic glycolysis would appear as a robust energy supply opposed to the more vulnerable mitochondrial bioenergetics.

## 4. The Oxygen Issue

In the blood the amount of glucose and oxygen (available from dissociation from hemoglobin) are of the same order of magnitude (3–5 mM). However, in the extracellular medium there are orders of magnitude between these two concentrations since the oxygen diffusion is driven by the concentration resulting from dissociation from hemoglobin. Therefore, this results in lower than 50 µM (0.05 mM) and measurements indicate a 20 µM concentration immediately outside the capillary [16]. Finally, oxygen concentration is likely to be in the low micromolar range at the level of mitochondria [17]. This contrasts with a more intense flux of oxygen than of substrates (one glucose, six O_2_). Therefore, the more common bioenergetics impairment in the mammalian organism originates from oxygen shortage. It could be the result of deterioration of vasculature (clot, trauma, inflammation) and/or of hypermetabolism (exercise, cancer, inflammation) making the possible O_2_ supply lower than required to feed cellular bioenergetics.

The efficiency of oxygen with regard to ATP production is quantified by the ATP/O_2_ ratio (Figure 1, Appendix A: oxphos). This ATP/O_2_ is influenced by the substrate oxidized due to different contribution of substrate linked phosphorylation steps and of different sites for electron entry into the mitochondrial respiratory chain (Appendix A). The ATP/O_2_ for the full oxidation of glucose is 34/6 = 5.7 and is considered as the reference in Figure 1. This value is high due to the ATP generation steps during glycolysis and the high ratio for reduction into NADH with regard to FAD/FMN steps (ten NADH versus two succinate dehydrogenase (*complex II*) reactions). The oxidation of palmitate takes place with a value close to five (4.96). Truncation of oxidative metabolism increases ATP/O_2_, with a value of 6.4 for glucose to citrate or succinate. The highest value is obtained with alphaketoglutarate (αKG) to succinate (Appendix A, ATP/O_2_ = 7.4). Oxidation of succinate is to be avoided, due to the poor ATP/O_2_ value of the succinate dehydrogenase step, compare in Figure 1 αKG-s versus αKG-a. Alphaketoglutarate could result from deamination of glutamine, which in contrast with the former is a quantitatively relevant substrate and is associated to metabolic adaptations in cancer, or inflammation [18]. Succinate, citrate/aconitate release has been observed under conditions of respiratory impairment [19,20] and/or inflammation [18]. While accumulation of these compounds may reflect the requirement for an increase in ATP/O_2_ for the aerobic pathway, the gain for the efficiency of oxygen remains modest in comparison to that resulting from a contribution of an anaerobic pathway that could increase indefinitely the overall ATP/O_2_. This is shown for an increasing contribution of lactic fermentation (Appendix A and Figure 1), which causes a sharp increase in glucose consumption (Figure 1 black curve) and for which twice higher release in lactic acid (not shown in Figure 1) is to be assumed.

As a consequence, if glucose is the cellular energy substrate and oxygen supply authorizes mitochondrial oxidation to cover 85% of ATP turnover the compensation for the remaining 15% by lactic fermentation multiplies by three glucose consumption with the result that lactic acid release and oxygen consumption rates are equal (inset in Appendix A). If respiration and lactic fermentation contribute equally to cellular bioenergetics (X = 2 on Figure 1) the rate of lactic acid release is 5.7 times higher than that of oxygen consumption (Inset in Appendix A). The same figures would result from any other factor other than oxygen limitation influencing the balance between glucose oxidation and lactic fermentation such as impairment of the pyruvate dehydrogenase (PDH) reaction. Therefore, comparison of lactate and oxygen fluxes does not provide a faithful image of their relative contribution to cellular bioenergetics and on the ground of lactate release the “Warburg effect” which might be observed although oxidative metabolism would, by far, remain the largest contributor to cellular bioenergetics. The growth of a tumor or inflammation induce hypermetabolism in the context of an altered and suboptimal vascularization, and both concur to make the ATP/O_2_ a major issue. Both cancer and innate immune response (inflammation) are associated to anaerobic energy production [21]. In addition, heterogeneity of tissue O_2_ concentration (Krogh model) is supposed to generate some lactate releasing domains and this even in absence of inflammation or cancer, this is reviewed in [22]. Finally, it should be noted that the formulation of Warburg effect as “lactate release although oxygen is sufficient” means actually “although oxygen is sufficient to ensure a better yield in ATP per glucose used”. This states implicitly that the main driver for metabolism would be the yield per glucose (substrate) before any other consideration, which is probably not always true.

## 5. Anoxic Mitochondrial Bioenergetics

An alternative strategy to lactic fermentation of glucose would be to use the oxphos machinery with the constraint that electrons should reduce another final acceptor than oxygen. Firstly, this would prevent reversion of mitochondrial bioenergetics that would consume glycolytic ATP to maintain mitochondrial membrane potential. Secondly, it has the advantage that substrates other than glucose could be used to sustain ATP regeneration.

### 5.1. Generation of Succinate by Reversion of Complex II

Strictly anaerobic mitochondrial bioenergetics has been shown to take place through mitochondrial *complex I* associated to the reoxidation of quinone by the mitochondrial *complex II* (succinate dehydrogenase) working in reverse mode using fumarate as the electron acceptor and releasing succinate (Figure 2), for a recent report in mammals see [19].

With the accepted stoichiometry between proton pumping and ATP it means 1.08 ATP for the four protons pumped by each *complex I* reaction. The ratio between succinate release and ATP formation is therefore close to one, hence similar to the lactate/ATP ratio of lactic fermentation. This requires intense fumarate supply, and hence reversion of the reactions of the TCA cycle from malate or oxaloacetate (Appendix A), which could be obtained from amino acids or by CO_2_ assimilation using pyruvate ATP and/or NADH provided by glycolysis. Three enzymes could be involved: pyruvate carboxylase (PC), phosphoenolpyruvate carboxykinase (PEPCK), or malic enzyme (ME) [23], this requires the reversion of the normal ME or PEPCK reaction (Appendix A). Since the role of phosphoenolpyruvate (PEP) was essentially considered, a “PEP metabolic branchpoint” was proposed [23] leading to anaerobic ATP production in invertebrates with succinate and alanine accumulation. It is noticeable that human deficiency in the carbon dioxide assimilating enzyme pyruvate carboxylase results in severe neonatal lactic acidosis [24]. While many mechanisms may explain this, it indicates that CO_2_ assimilating reaction ought to take place at a significant rate. Then, considering a combined glucose and glutamine metabolism (Appendix A), converging to succinate improves ATP/O_2_, and uses significantly less glucose than the equivalent combination of glucose oxidation and lactic fermentation (Figure 1, Appendix A). A consequence would be CO_2_ incorporation replenishing intermediates of the Krebs cycle a process known as anaplerosis.

### 5.2. High Requirement for Complex I and II Activities

This anoxic mode for the mitochondrial respiratory chain shows different requirements with regard to respiratory complexes activities when compared to the aerobic pathway. The complete oxidation of glucose into CO_2_ regenerates 4 ATP, releases 10 NADH, and requires two *complex II* reactions. The other 30 ATP result from the oxidative phosphorylation pathway (with a supposed yield of 100%) and therefore imply 30 reactions of phosphorylation by *complex V*. The number of reactions by other mitochondrial complexes could be enumerated: in addition to the two *complex II* reactions the ten NADH would cause the same number of reactions by *complex I*. The result is twelve entries of electrons in the mitochondrial respiratory chain and reactions of the *complexes III* and *IV*.

In contrast, the anaerobic pathway does not require *complex III* and *IV* reactions and the number of reactions is the same for *complex I* and *complex II* (Figure 2 bottom). For the same number of ATP (*complex V* reactions) 30/1.08 ≈ 28 redox reactions in *complexes I* and *II* are required. This means nearly three times (*complex I*) or 14 times (*complex II*) more than in aerobic conditions. This may have consequences for cells using this anaerobic mode of the mitochondrial respiratory chain: First, the intense requirement for *complex I* and *II* activities could result in a much higher sensitivity to impairment of *complex I* or *II* by mutation or intoxication. Second, in cells adapted to recourse to this anaerobic pathway the ratio of enzymatic activities between *complexes I*, *II* and the others (*III*–*V*) is expected to be altered in comparison with strictly aerobic cell types. Notably, examination of ratios between the activities of the different complexes evidenced such differences with the brain characterized by relative over-representation of *complexes I* and *II*, when compared to *complex V* [25].

## 6. Time and Space

Lactic acid or succinate release may correspond to a permanent anaerobic lifestyle, which is restricted to a minority of animal species. However, in the vast majority anaerobic metabolism results from transient imbalance between oxygen supply and needs, such as during ischemic shock or intense stimulation/exercise. Lactate and succinate accumulation built up a metabolic and oxygen debt reimbursed later by mitochondrial oxidative metabolism when oxygen becomes available. This time-based relationship finds an echo in spatial organization and muscles/erythrocytes and liver (in the long range) [22,26], glial cells and neurons [27,28] or stromal and cancer cells [29] (in the short range) constitutes examples of spatially organized metabolic synergy between lactate producers and lactate consumers. Similar spatial organization with succinate appears to emerge in the retina [30]. Then the anaerobic metabolism based on succinate generation described long time ago in invertebrates [23] is now recognized in mammals [30] suggesting it as a general strategy.

### 6.1. Succinate Reoxidation and ROS Release Proximal to Hypoxic Domain

Upon reperfusion (reoxygenation) the succinate accumulated is intensely oxidized by *complex II* [19], which causes intense electron supply to respiratory chain. Two factors would explain this absolute priority for succinate consumption: (1) The very same enzyme succinate dehydrogenase (*complex II*) ensures either building of the succinate oxygen debt or electron injection in the respiratory chain. In comparison, the pathway from lactic acid to electron supply to the respiratory chain requires much more steps [26]; (2) Since both *complex I* and *complex II* aim to reduce the quinone (Figure 2 top) the intense *complex II* activity impairs the forward reaction by *complex I* (NADH oxidation) and at the opposite end promotes the reverse reaction (reduction of NAD), hence inverse reactions of that shown at the bottom part of Figure 2. This has two consequences: the first is to promote oxidative stress [19] since reversion of *complex I* increases greatly superoxide release. The second is that it impairs contribution of *complex I* to oxidative phosphorylation and to further oxidation of the fumarate released by *complex II* reaction. Therefore, it results in a prominent (if not exclusive) contribution of *complex II* to oxidative phosphorylation with the theoretical value of 1.6 for the ATP/succinate and ATP/O ratios. In contrast, full lactate oxidation takes place with large contribution of *complex I*, and much higher yield (ATP/lactate = 16).

The consequences could be understood by considering the situation in which the metabolism of a single cell is fully anaerobic and releases either lactate or succinate, which is oxidized by neighboring fully aerobic oxidative cells. The generation of 100 ATP by lactic fermentation releases 100 lactic acid molecules, and their full oxidation would release 100 × 16 = 1600 ATP hence enough to sustain the same ATP generation in sixteen cells. If anaerobic succinate generation as shown in Figure 2 is considered it results in 1.08 ATP/succinate hence 100/1.08 ≈ 93 succinate molecules are generated. Then with the figures above the partial oxidation of the same number of succinate molecules by *complex II* with exclusion of *complex I* reaction would release 93 × 1.6 = 149 ATP, and hence two cells would be more than enough to eliminate all of this succinate. Therefore, while lactate may diffuse away from the emitting cells the succinate would be eliminated proximal to its origin. Another difference is the requirement in oxygen, full oxidation of lactate takes place with an ATP/O_2_ ratio of 5.4. Hence if glucose oxidation is taken as a reference ATP/O_2_ = 5.7 there is a 6% increase in oxygen consumption caused by the shift from glucose to lactate (5.7/5.4 = 1.06). In comparison, the partial oxidation of succinate by *complex II* takes place with consumption of one oxygen atom and leads to the formation of 1.6 ATP, and hence an ATP/O_2_ of 3.2 (Figure 2). Then with reference to glucose the increase in oxygen consumption would be 78% (5.7/3.2 = 1.78). This is shown in the Figure 1 by the open cycle at the upper end of the dotted part of the oxygen consumption curve. Consequently, while lactate full oxidation feeds a large number of cells in which the oxygen consumption is marginally increased, the fast and partial succinate reoxidation would feed few cells in which oxygen consumption is greatly increased.

The fate of the fumarate generated by the *complex II* during this fast and exclusive reoxidation of succinate remains to be examined. Whether fumarate is released by the succinate oxidizing cells is unknown. Theoretically, the reversion of the reactions from pyruvate to fumarate (Appendix A) would be possible (Appendix A). If reoxidation of NADH by *complex I* is excluded the option would be malate or lactate (Appendix A) hence ME or PEPCK would withdraw TCA intermediates (cataplerosis), a role recognized for PEPCK [31], and cancel the anaplerosis associated to the anaerobic succinate metabolism (see Section 5.1). Then, lactate could be the final product of a “succinate cycle” associating anaerobic succinate generation to its proximal, fast, partial and low yield reoxidation. Then, this succinate cycle may well occur at a significant rate but the succinate involved never reach the general circulation and thus remain undetected. This succinate cycle may also explain the counterintuitive release of reactive oxygen species (ROS) associated to hypoxia. These ROS would originate from intense succinate reoxidation in the oxygenated periphery of the hypoxic region. While the hypoxic core would be likely to cause massive cell death the peripheric succinate oxidation area is likely to constitute a source of survivor cells unfortunately exposed to ROS in the aggravating context of a deteriorated cellular bioenergetics [32].

### 6.2. A Succinate Barrier to Oxygen Diffusion

The low efficiency of partial oxidation of succinate with regard to oxygen (see above) results in a much more intense oxygen consumption by mitochondria at the border of the hypoxic region. Consequently, the aerobic mitochondria oxidizing succinate would build a barrier against oxygen diffusion towards the anoxic mitochondria releasing it. On one side, this constitutes an aggravating factor stabilizing the hypoxic domain, but on the other side it may be used for the protection of oxygen sensitive cellular structures. The relevance of intracellular oxygen gradients is debated [33,34]. Indirect support could be found in experimental protocols used in functional studies with nuclei or mitochondria. Nuclear biochemical activities (transcription, splicing) requires the presence of millimolar concentrations of the reducing agents dithiothreitol or sodium bisulfite [35,36]. In contrast, mitochondrial preparation and functional tests take place in the presence of air saturated media, and hence with oxygen concentration orders of magnitude higher than intracellular values. This is with little deterioration of their performance, although oxidative damage could be shown to take place with time [37].

The PEP metabolic branch point (see Section 5.1) would cause anaplerosis or not according to oxygen concentration and this within a single cell. The hypoxic metabolites (lactic acid, citrate, succinate, alanine), are therefore expected to stimulate metabolism in two ways: to reimburse the oxygen debt but also by stimulation of biosynthesis (Figure 3) and cell division.

## 7. Repression of Respiration

We would like to end by advocating that the Warburg effect represents a defense to prevent the hypoxic conditions to occur, and therefore to prevent their deleterious consequences. Glucose and oxygen are provided to cells by capillaries, which raises the issue of inequality of cells with regard to this supply. Then, one should consider what glucose and oxygen share between different cell layers either close or remote from the oxygen and glucose source (capillary). The results are shown in a model (Figure 4) with three successive cell layers associated to a capillary delivering a constant blood flow defining a quantity of glucose and oxygen available. In addition, the oxygen supply is supposed to be insufficient to allow full respiratory activity in the three layers. There is no obvious mechanism to ensure the same optimal share between oxidative phosphorylation and lactic fermentation in the different layers (Figure 4B). Then, two schemes for inequal use are explored: (1) mitochondrial respiration shows priority and due to its high affinity takes place at a same rate as long as oxygen is present (Figure 4C); (2) alternatively lactic fermentation takes place first and is considered as proportionate to glucose concentration (Figure 4D). This last proposal equalizes the use of glucose and oxygen and fits better with diffusion constraints since gases (O_2_, CO_2_) move faster than glucose or lactic acid. Notably, endothelial cells are closest to oxygen and glucose supply and rely heavily on aerobic glycolysis [38]. Building of the model “D” in the Figure 4 required the proportionality between glucose concentration and aerobic glycolysis intensity and its precedence over respiration. In agreement with these requirements the present knowledge indicates that the glycolytic intermediate Fructose 1,6 biphosphate (F1,6BP) is an inhibitor of yeast or mammalian mitochondrial respiration and exerts its effect at the level of mitochondrial respiratory complexes [39]. This observation was proposed as a mechanistic explanation for the “Crabtree effect” which refers to an immediate partial repression of mitochondrial respiration after abrupt increase in the concentration of glucose although oxygen supply is unchanged [40].

## 8. Conclusions

The aim of this paper is to attract attention to the fact that the Warburg effect cannot be considered only on the ground of its deteriorated yield with regard to conversion of glucose into ATP, but that many other criteria must be considered to evaluate its value with regard to cellular bioenergetics. For example, relatively simple models could explain the Warburg effect and glutamine use by the need to increase the yield of oxygen use (ratio ATP/O_2_) to feed cellular ATP turnover. The existence of a genuine Warburg effect could be questioned when lactic acid reveals actually mitochondrial oxphos impairment and not a metabolic preference for the low yield aerobic glycolysis (Appendix A). The consequences of the metabolic alterations increasing ATP/O_2_ diminish/exclude complete oxidation of substrates into CO_2_ and at the opposite may lead to CO_2_ assimilation with the release of organic molecules (lactic acid, citrate, succinate), which may constitute a signal promoting illegitimate biosynthesis and cell division in a mutagenic context. Transient ischemia constitutes an acute inducer of this process, and hypermetabolism and vasculature deterioration linked to chronic inflammation may constitute a long-term driver for this “at risk” energy metabolism, which would continue during tumor growth.

## Figures and Tables

**Figure 1 biology-10-01000-f001:**
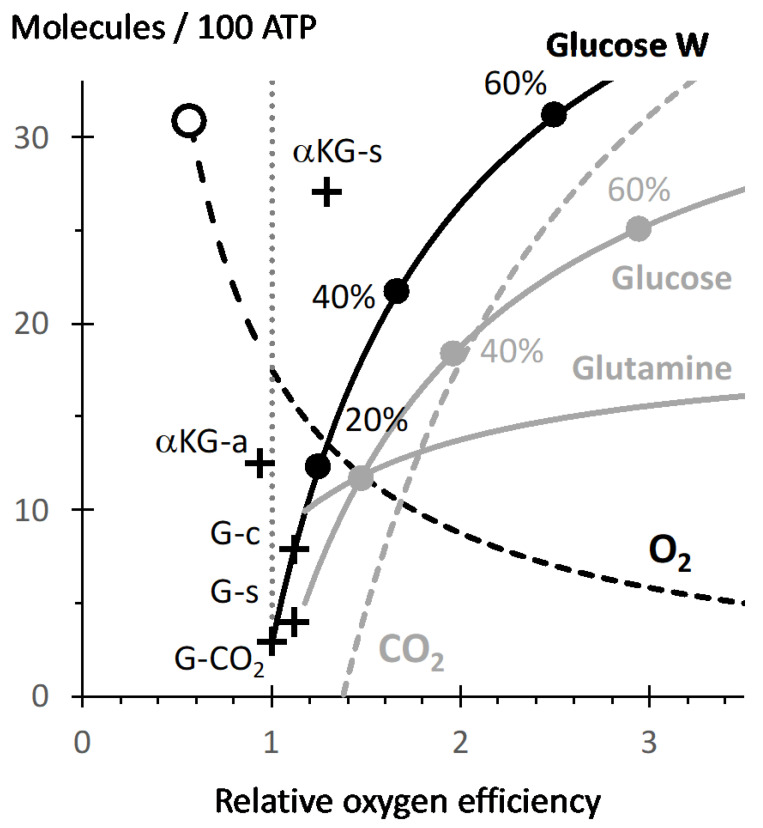
The *X*-axis indicates the ATP/O_2_ ratio expressed as relative to that of full oxidation of glucose into CO_2_ (actual value of ATP/O_2_ = 5.7), a vertical dotted line highlights this reference value. The *Y*-axis represents the number of the different substrates required for 100 phosphorylation reactions generating ATP. Crosses represent these values for different oxidative pathways: glucose to CO_2_ (G-CO_2,_ which uses 2.9 glucose), glucose to succinate (G-s) or to citrate (G-c), alphaketoglutarate to oxaloacetate (aKG-a) or to succinate (aKG-s). The black curve (Glucose W) starts from X = 1 and Y = 2.9 (see above) and relates the evolution for glucose utilization as contribution of lactic fermentation increases, dots represent successive increases by 20%. Grey curves represent the consumption of glucose and glutamine releasing alanine and succinate with the aerobic/anaerobic pathway presented in Appendix A. The dashed grey curve represents the CO_2_ flux in this same pathway but with inverse ordinates: it becomes positive when net fixation of CO_2_ occurs (X = 1.4). The dashed black curve (hyperbolic) represents the oxygen consumption. The longer dashes, indicate values starting from oxidation of succinate into malate SDH reaction (empty dot X = 0.57) to that of glucose (X = 1, Y = 17.5O_2_) and therefore represent the range of values for oxygen consumption expected from reoxidation of succinate when accumulated (see text).

**Figure 2 biology-10-01000-f002:**
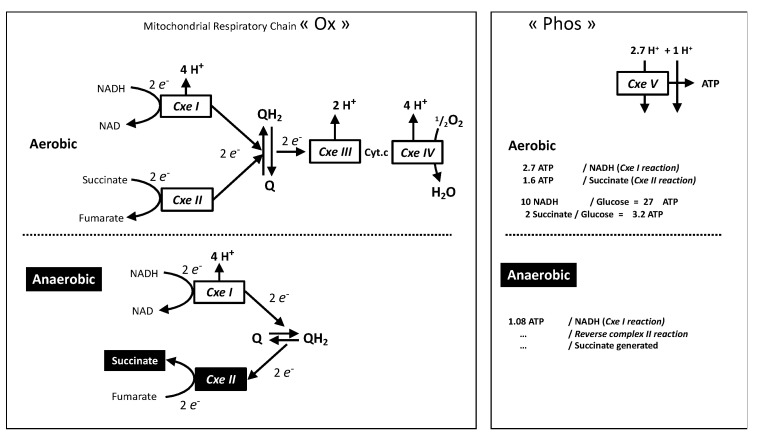
The oxidative phosphorylation machinery is split into “Ox”: *complexes I*–*IV* and “Phos”: *complex V* with ADP/ATP and Phosphate transporters (not shown). Redox intermediates in the respiratory chain are quinone (Q/QH2) and cytochrome c (Cyt.c). Stoichiometric relationship between proton movement and reactions catalyzed by the complexes are shown, this refers to the number of protons pumped per two electrons (Ox) or to proton-return reactions (Phos) required for generation of one cellular ATP by *complex V* and exchange of ions (ADP, ATP, Pi) between mitochondrial and cytosol. The aerobic situation “normal cellular bioenergetics” (**top**) is compared to the succinate generating anaerobic pathway (**bottom**).

**Figure 3 biology-10-01000-f003:**
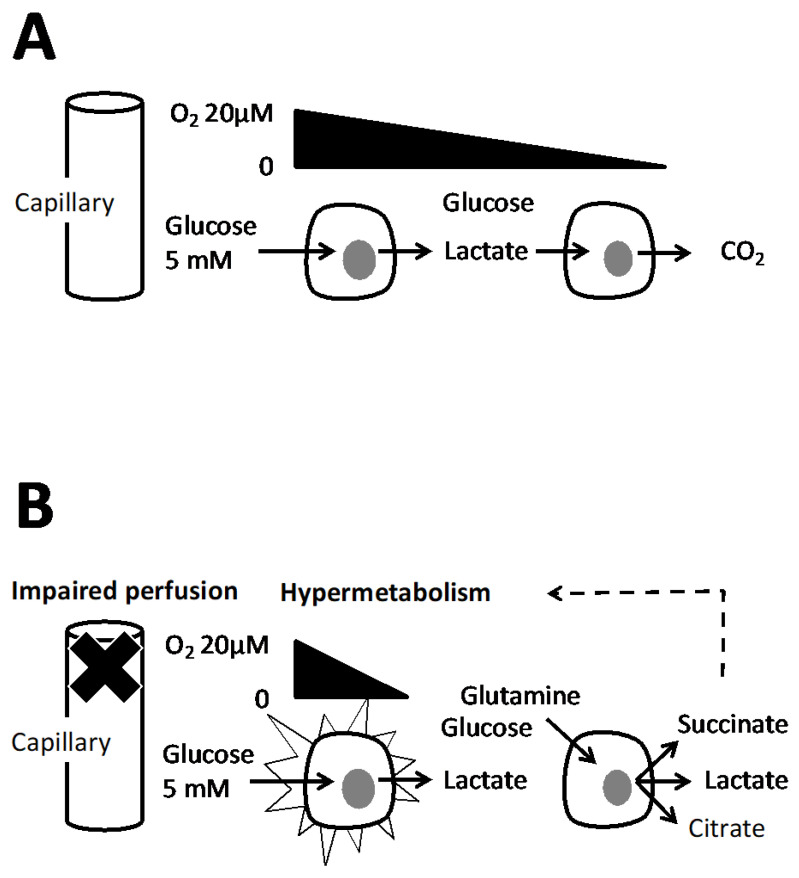
A capillary feeds different layers of cells schematized as two cells with a different distance to capillary. The flux of oxygen and of wastes are figured by arrows, (**A**) the oxygen supply is sufficient however Warburg effect may take place (see Section 7) but lactate is eventually oxidized in CO_2_. (**B**) Alteration of vasculature and/or hypermetabolism results in an oxygen supply that could not cover all cellular needs. Then metabolic adaptations take place in the second cell (see text) with the result of release of lactate, citrate and succinate. They trigger hypermetabolism to reimburse the oxygen debt or stimulate biosynthesis. The dotted arrow suggests that these effects could be exerted locally.

**Figure 4 biology-10-01000-f004:**
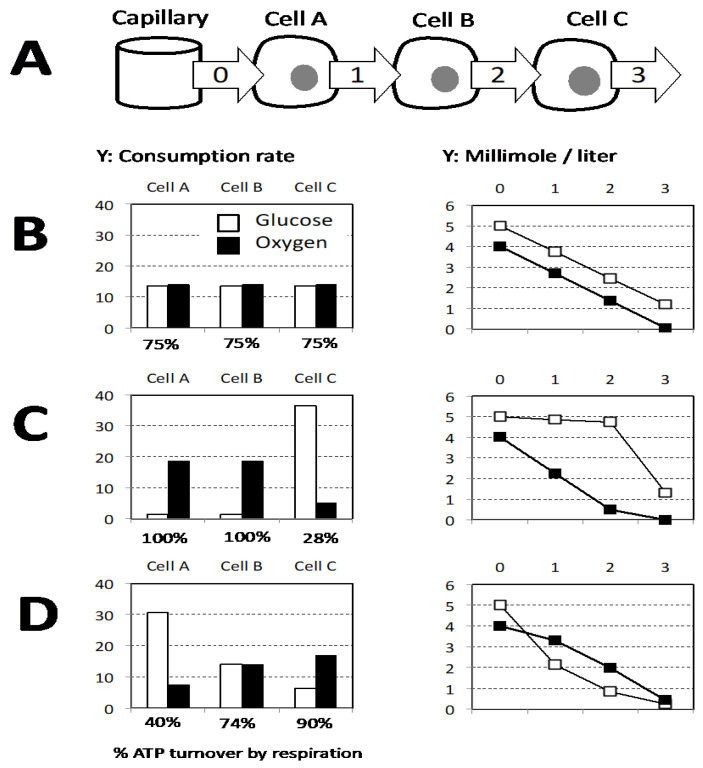
(**A**) The capillary supplies oxygen and glucose and three successive layers of cells with increasing distance from capillary are considered (Cell A, B, C), relevant extracellular sites are figured by arrows that schematize the flux from the capillary to the successive layer of cells. Glucose and oxygen available in the blood are considered to be present immediately outside the capillary (site 0). In the successive sites (1–3) the quantities available are supposed to be determined by the consumption of the previous cell layer. The oxygen supply is supposed to allow a mitochondrial respiration covering 75% of the sum of the cellular ATP needs. (**B**–**D**) The histograms on the left figure glucose and oxygen consumption rate of each cell layer, the graphs on the right represent the concentrations available in the successive sites (0–3). (**B**) Equal share of oxygen and glucose for the three layers results in a linear decrease of concentrations. (**C**) Respiration is the priority, and then the first two cells (A and B) cover 100% of their ATP need by respiration and the lactic fermentation is restricted to the remote cell C. (**D**) Lactic fermentation is the priority and is considered as proportionate to glucose concentration with a value arbitrarily set to Lactic fermentation flux = 12× glucose concentration.

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
