# Peer review of "Warburg Effect, Glutamine, Succinate, Alanine, When Oxygen Matters"

_biology, 2021, doi:10.3390/biology10101000_

Round 1
Reviewer 1 Report
The authors described the consequences of Warburg effect on metabolic pathways particularly focusing on bioenergetics.
The review sounds to be interesting.However, before to consider this manuscript suitable for publication, the following points should be addressed by authors.
- Add “Warburg effect” in the title of the manuscript.
- Include a summary table in the conclusions illustrating the different actions of Warburg effect on anabolic processes.
Author Response
Authors:
Reviewer:
The authors described the consequences of Warburg effect on metabolic pathways particularly focusing on bioenergetics.
The review sounds to be interesting. However, before to consider this manuscript suitable for publication, the following points should be addressed by authors.
Authors:
Reviewer:
- Add “Warburg effect” in the title of the manuscript.
Authors:
This is done, present title: “Warburg effect, glutamine, succinate, alanine, when oxygen matters”
Reviewer:
- Include a summary table in the conclusions illustrating the different actions of Warburg effect on anabolic processes.
Authors:
We understand this demand for a “graphic summary” that could be in the form of a table. However, it seems to us that it not possible to restrict this to the impact of Warburg effect on anabolic processes for several reasons 1) Warburg effect, as such seems, to us rather a competitor than a promoter of anabolism (§ 2). 2) Anabolic processes are mentioned independently of the Warburg effect and in relation with anaerobic metabolism, and with reference to other mechanism than Warburg effect (§.
We propose a decision flowchart (Figure S7), which summarizes the criteria for recognition of a genuine Warburg effect, and, with reference to chapters of the text. In addition, we propose a simplified version as a graphical abstract.
One more modification
We include an additional Author Dr Noureddine Hammad. He participated to earliest discussions about Warburg effect, and for this reason his name was included in the earliest manuscript project. However due to negligence his name disappeared in subsequent draft at the origin of the present opinion paper. We respectfully request the acceptance of what may be seen as a late modification of authorship but is in fact the restoration of the initial state.
Reviewer 2 Report
This is a highly interesting article that deserves a broad readership beyond biochemisty and molecular biologist, i.e. oncologist and molecular pathologist like this reviewer. However, the current text is extremely complicated and requires extreme pre-knowledge to follow the authors’ arguments - to the degree that it might deter many potential readers from reading. Therefore, it is a general recommendation to explain in a way that arguments become more transparent and better to grasp.
Accordingly, one recommndation is to introduce a glossary to explain terms that may be trivial to biochemisty but not to people like me who have been in the cancer field for decades
Strikingly, the language needs significant improvement particularly in the beginning, but would profit fom correction through a native speaker. Punctuation is often missing, making reading very difficult
Details:
Fig. S1 Legends should be rather self-explanatory! Here, many abbreviations are not explained. Meaning of +2 etc. not clear. Meaning e.g. of F1,6BP (Fructose 1,6 biphosphate – explained later) is not given.. Different colors of arrows remains unclear
Fig. S2 Abbreviations are not explained e.g. OAA; terms such as “engagement” should be explained (>glossary)
Legend to Fig. S2 not clear to this reviewer – please try to be more transparent: “For calculation of the Oxphos ATP generation the number of protons pumped by NADH (10) or FMN/FAD/Succinate (6) oxidation in the respiratory chain is considered and the conversion into ATP uses the theoretical stoichiometry of 3.7 H+ per ATP..”
Also unclear: “For example, for the reaction aKG to succinate the value is 4.4 (Y axis) but with one CO2 out of the five carbons in aKG the value rises to 22 (five times more) if only CO2 release is considered.”
Line 188 unclear how the authors arrive at a 5.7 times higher lactate production
Lines 201-204 Unclear to this reviewr:…In other terms the formulation of Warburgeffect as 1) “lactate release” and 2) “although oxygen is sufficient” implies stringent requirements with regard to this second assertion and assumes that the criterion to be considered (driver for energy metabolism) is the yield of ATP per substrate before any other constraint (see above). Please try rewording that is clearer
Fig. S3 Abbreviation of alanine (Ala) should be added to the legend
Page 6 “Anaplerosis” should be explained > Glossary
Page 7 Lactic acid or of succinate release … > should correctly read: Lactic acid or succinate release…
Line 286. Unclear to this reviewer: “Because complex I and complex II share the same electron acceptor (Figure 2 top)…”
Line 297-298 “As a consequence, intense succinate consumption takes place proximal to the anaerobic domain while lactate could travel over longer distances”. Last part of the sentence: “…while lactate could travel…” not clear to this reviewer.
It is also unclear, why the subsequent pargraph (Line 299) starts with “Firstly,…”
Line 322 Please explain “PEP” in “The PEP metabolic branchpoint… (Sorry – where is it mentioned above? In fact it occur in Fig. S1 but is not explained there either)
Line 359 Unclear conclusion: “...This requires a mechanism by which aerobic glycolysis would take over respiration independently from oxygen presence/availability.”
The Figures S4 and S5 appear are not mentioned in the text (only at the very end without link to statements in the text). Please refer to the Figures inside the text or delete.
Author Response
Reviewer:
This is a highly interesting article that deserves a broad readership beyond biochemisty and molecular biologist, i.e. oncologist and molecular pathologist like this reviewer. However, the current text is extremely complicated and requires extreme pre-knowledge to follow the authors’ arguments - to the degree that it might deter many potential readers from reading. Therefore, it is a general recommendation to explain in a way that arguments become more transparent and better to grasp.
Authors:
Thank you very much for this positive comment we are particularly sensitive to it because that is exactly the purpose of the present manuscript. We understand the difficulties experienced by the reviewer; therefore, we hope that the revised manuscript improves understanding by people outside the metabolic/bioenergetic field.
Reviewer:
Accordingly, one recommandation is to introduce a glossary to explain terms that may be trivial to biochemisty but not to people like me who have been in the cancer field for decades
Authors:
We tried to address this issue in two ways. First, more terms would be explained in the text. Second, as requested, we provide the supplementary information “S8: Glossary and abbreviations”, which recapitulate the meaning of all abbreviations used and provides a quick explanation of several biochemical/bioenergetical principles of use here.
Reviewer:
Strikingly, the language needs significant improvement particularly in the beginning, but would profit fom correction through a native speaker. Punctuation is often missing, making reading very difficult
Authors:
We tried to improve this, an external reader from the US went through the manuscript and suggested modifications. We expect that page charges include further correction of the accepted manuscript
Details:
Reviewer:
Fig. S1 Legends should be rather self-explanatory! Here, many abbreviations are not explained. Meaning of +2 etc. not clear. Meaning e.g. of F1,6BP (Fructose 1,6 biphosphate – explained later) is not given. Different colors of arrows remain unclear
Authors:
Figure 1 has been modified to improve the use of colors, notably in the previous version the orange color was used with two different meanings: C6 intermediates of glycolysis or negative ATP balance; while both overlap to a significant extent, they are not exactly coincident. The range for C6 intermediates is now represented in blue. Explanation of the abbreviations is now given.
Reviewer:
Fig. S2 Abbreviations are not explained e.g. OAA; terms such as “engagement” should be explained (>glossary)
Authors:
The term OAA is now explained in the legend of Figure S2, in the text and recalled in S8. Engagement has been replaced by “mobilization” or “use” (we hope it is better).
Reviewer:
Legend to Fig. S2 not clear to this reviewer – please try to be more transparent: “For calculation of the Oxphos ATP generation the number of protons pumped by NADH (10) or FMN/FAD/Succinate (6) oxidation in the respiratory chain is considered and the conversion into ATP uses the theoretical stoichiometry of 3.7 H+ per ATP..”
Authors:
Calculation for ATP generation by oxidative phosphorylation is explained in a shorter way, with no reference to proton pumping, useless here; moreover, detailed explanations are now provided in S8.
Now:
Calculation of the ATP generation by the oxidative phosphorylation is made as follow: re-oxidation of NADH is supposed to be in all cases mediated by entry of electrons at the level of the complex I of the mitochondrial respiratory chain and to yield 2.7 ATP per oxygen atom (ATP/O2=5.4). If the electrons come from a FAD/FMN intermediate, this includes Complex II of the mitochondrial respiratory chain,the yield is considered to be 1.6 ATP per oxygen atom (ATP/O2=3.2).
Reviewer:
Also unclear: “For example, for the reaction aKG to succinate the value is 4.4 (Y axis) but with one CO2 out of the five carbons in aKG the value rises to 22 (five times more) if only CO2 release is considered.”
Authors:
This “CO2” issue is now explained in more details.
Now:
Note that in the case of complete oxidation the yield per carbons of substrate and per CO2 released are the same but their values are obviously different if partial oxidation is considered. For example, if the pathway from aKG to succinate is considered it mobilizes five carbons (aKG), releases one CO2 and yields 3.7 ATP. The yield per six carbons of aKG is therefore 3.7×6/5= 4.4 (this figure). In contrast, with regard CO2 the comparison with values for complete oxidation would require to consider six CO2 hence six reactions and the value would be 3.7×6=22 hence closer to the values obtained with glucose or pyruvate.
Reviewer:
Line 188 unclear how the authors arrive at a 5.7 times higher lactate production
Authors:
As an explanation we provide an inset in the Figure S2 to highlight the quantitative consequences of a 15% or 50% contribution of aerobic glycolysis to cellular ATP generation.
Now (legend of inset Figure S2):
Inset: Lactate release (blue) and consumption of oxygen (red) or glucose (black) as the contribution of lactate fermentation to the cellular ATP turnover rate increases. The generation of 100 ATP molecules is considered, X is the number of ATP generated by lactic fermentation the rest is supposed to come from glucose oxidation, then with X=0 values from the full oxidation of glucose into CO2 are represented: no lactic acid generated (Y=0), approximately three glucose and 18 oxygen (O2) molecules are used bythe oxidation pathway. The numbers of glucose and oxygen used are also indicated for two remarkable situations: lactate and oxygen flux are equal in intensity (15% lactic ATP) or lactic fermentation and oxidation contribute equally to ATP generation (50% lactic ATP). In the former case the glucose flux is 10 (three times increase) and the oxygen flux is 15 (ATP/O2=100/15≈6.7), in the latter case the oxygen flux has a value of 8.8 and the lactate flux of 50 hence 5.7 times higher.
Reviewer:
Lines 201-204 Unclear to this reviewr:…In other terms the formulation of Warburgeffect as 1) “lactate release” and 2) “although oxygen is sufficient” implies stringent requirements with regard to this second assertion and assumes that the criterion to be considered (driver for energy metabolism) is the yield of ATP per substrate before any other constraint (see above). Please try rewording that is clearer
Authors: This part has been reformulated to:
Finally, it should be noted that the formulation of Warburg effect as “lactate release although oxygen is sufficient” means actually “although oxygen is sufficient to ensure a better yield in ATP per glucose used”. This states implicitly that the main driver for metabolism would be the yield per glucose (substrate) before any other consideration, which is probably not always true.
Reviewer:
Fig. S3 Abbreviation of alanine (Ala) should be added to the legend
Authors: Done
Reviewer:
Page 6 “Anaplerosis” should be explained > Glossary
Authors: This is done in the present version, in the text and in S8.
Now:
A consequence would be CO2 incorporation replenishing intermediates of the Krebs cycle a process known as anaplerosis.
Reviewer:
Page 7 Lactic acid or of succinate release … > should correctly read: Lactic acid or succinate release…
Authors: This is corrected
Reviewer:
Line 286. Unclear to this reviewer: “Because complex I and complex II share the same electron acceptor (Figure 2 top)…”
Authors: reformulated to:
“2) Because both complex I and complex II aim to reduce the quinone (Figure 2 top) the intense complex II activity impairs the forward reaction by complex I (NADH oxidation) … “
Reviewer:
Line 297-298 “As a consequence, intense succinate consumption takes place proximal to the anaerobic domain while lactate could travel over longer distances”. Last part of the sentence: “…while lactate could travel…” not clear to this reviewer.
Authors: OK, this deserves detailed, and therefore longer, explanations.
Now:.
… Therefore, it results in a prominent (if not exclusive) contribution of complex II to oxidative phosphorylation with the theoretical value of 1.6 for the ATP/succinate and ATP/O ratios. In contrast, full lactate oxidation takes place with large contribution of complex I, and much higher yield (ATP/lactate=16).
The consequences could be understood by considering the situation in which the metabolism of a single cell is fully anaerobic and releases either lactate or succinate, which is oxidized by neighboring fully aerobic oxidative cells. The generation of 100 ATP by lactic fermentation releases 100 lactic acid molecules, their full oxidation would release 100×16=1600 ATP hence enough to sustain the same ATP generation in sixteen cells. If anaerobic succinate generation as shown in Figure 2 is considered it results in 1.08 ATP/succinate hence 100/1.08≈93 succinate molecules are generated. Then with the figures above the partial oxidation of the same number of succinate molecules by complex II with exclusion of complex Ireaction would release 93×1.6=149ATP hence two cells would be more than enough to eliminate all of this succinate. Therefore, while lactate may diffuse away from the emitting cells the succinate would be eliminated proximal to its origin. Another difference is the requirement in oxygen, full oxidation of lactate takes place with an ATP/O2 ratio of 5.4. Hence if glucose oxidation is taken as a reference ATP/O2=5.7 there is a 6% increase in oxygen consumption caused by the shift from glucose to lactate (5.7/5.4=1.06). In comparison, the partial oxidation of succinate by complex II takes place with consumption of one oxygen atom and leads to the formation of 1.6 ATP hence an ATP/O2 of 3.2 (Figure 2). Then with reference to glucose the increase in oxygen consumption would be 78% (5.7/3.2=1.78). This is shown in the Figure 1 by the open cycle at the upper end of the dotted part of the oxygen consumption curve. Consequently, while lactate full oxidation feeds a large number of cells in which the oxygen consumption is marginally increased, the fast and partial succinate reoxidation would feed few cells in which oxygen consumption is greatly increased.
The fate of the fumarate generated by the complex II during this fast and exclusive reoxidation of succinate remains to be examined. Whether fumarate is released by the succinate oxidizing cells is unknown. Theoretically, the reversion of the reactions from pyruvate to fumarate (Figure S6) would be possible (Figure S3). If reoxidation of NADH by complex I reaction is to be excluded the option would be malate or lactate (Figure S3B) hence ME or PEPCK would withdraw TCA intermediates (cataplerosis) a role recognized for PEPCK [31], and with regard to TCA intermediates cancel the anaplerosis associated to the anaerobic succinate metabolism (see 5.1). Then, lactate could be the final product of a “succinate cycle” associating anaerobic succinate generation to its proximal, fast, partial and low yield reoxidation. Then, this succinate cycle may well occur at a significant rate but the succinate involved never reach the general circulation and thus remain undetected. …
Reviewer:
It is also unclear, why the subsequent paragraph (Line 299) starts with “Firstly,…”
Authors: the text has been modified see above
Reviewer:
Line 322 Please explain “PEP” in “The PEP metabolic branchpoint… (Sorry – where is it mentioned above? In fact it occur in Fig. S1 but is not explained there either)
Authors: This PEP branchpoint is now presented in greater detail I the lines 233-235, we introduced a Figure S3 to illustrate the metabolic network at stake here. This Figure S3 is also used to explain the succinate to lactate issue (Figure S3B).
Now:
… This requires intense fumarate supply, hence reversion of the reactions of the TCA cycle from malate or oxaloacetate (Figure S3), which could be obtained from amino acids or by CO2 assimilation using pyruvate ATP and/or NADH provided by glycolysis. Three enzymes could be involved: Pyruvate carboxylase (PC), phosphoenolpyruvate carboxykinase (PEPCK) or malic enzyme (ME) [23], this requires the reversion of the normal ME or PEPCK reaction (Figure S3). Because the role of phosphoenolpyruvate (PEP) was essentially considered a “PEP metabolic branchpoint” was proposed [23] leading to anaerobic ATP production in invertebrates with succinate and alanine accumulation...
Later on line 371 (former 322)
The PEP metabolic branchpoint (see above 5.1)…
Reviewer:
Line 359 Unclear conclusion: “...This requires a mechanism by which aerobic glycolysis would take over respiration independently from oxygen presence/availability.”
Authors: modified to:
… Building of the model “D” in the figure 4 required the proportionality between glucose concentration and aerobic glycolysis intensity and its precedence over respiration. In agreement with these requirements the present knowledge indicates that the glycolytic intermediate Fructose 1,6 biphosphate (F1,6BP) is an inhibitor of yeast or mammalian mitochondrial respiration and exerts its effect at the level of mitochondrial respiratory complexes [39]. This observation was proposed as a mechanistic explanation for the “Crabtree effect” which refers to an immediate partial repression of mitochondrial respiration after abrupt increase in the concentration of glucose although oxygen supply is unchanged….
Reviewer:
The Figures S4 and S5 appear are not mentioned in the text (only at the very end without link to statements in the text). Please refer to the Figures inside the text or delete.
Authors: As we consider that the length of supplementary materials is not a too serious issue, we consider preferable to retain these Figures in the revised version. They explain better the different metabolic circuit in aerobic or anaerobic use of glutamine. They are also mentioned when relevant line 240-243
… Then considering a combined glucose and glutamine metabolism (Figures S3-S5) converging to succinate would engage significantly less glucose than the equivalent combination of glucose oxidation and lactic fermentation (Figure 1, Figures S4, S5)…
Authors: One more modification
We include an additional Author Dr Noureddine Hammad. He participated to earliest discussions about Warburg effect, and for this reason his name was included in the earliest manuscript project. However due to negligence his name disappeared in subsequent draft at the origin of the present opinion paper. We respectfully request the acceptance of what may be seen as a late modification of authorship but is in fact the restoration of the initial state.
Reviewer 3 Report
The two authors, Dr. Bouillaud and Dr. Schwartz, present a very extensive and detailed manuscript on the subject of the Warburg Effect and its metabolic interfaces and consequences. 'Opinion' was chosen as the manuscript type, which I very much support, as in some cases personal, scientific hypotheses are conveyed. I am convinced of the structure of the manuscript.
What I find very difficult is partly the structure of the sentence and the English expression. It is sometimes very difficult for the reader to follow the content. This is very often due to the fact that the sentences are formulated too awkwardly and commas are often missing or not correctly set. I recommend a complete overhaul of the English language. In my opinion, the manuscript will then have the quality to be accepted in the journal "Biology".
There are some spelling mistakes, for example a space is always placed between number and unit (line 133,. But also with the hyphen (line 130) and the and equal sign (line 180). Please carefully revise the manuscript with regard to these small spelling mistakes!
I think the supplementary figures are worth showing. If at all possible I would include part of the supplementary figures in the main manuscript.
Author Response
Reviewer:
The two authors, Dr. Bouillaud and Dr. Schwartz, present a very extensive and detailed manuscript on the subject of the Warburg Effect and its metabolic interfaces and consequences. 'Opinion' was chosen as the manuscript type, which I very much support, as in some cases personal, scientific hypotheses are conveyed. I am convinced of the structure of the manuscript.
Authors:
Thank you for your positive comments.
Reviewer:
What I find very difficult is partly the structure of the sentence and the English expression. It is sometimes very difficult for the reader to follow the content. This is very often due to the fact that the sentences are formulated too awkwardly and commas are often missing or not correctly set. I recommend a complete overhaul of the English language. In my opinion, the manuscript will then have the quality to be accepted in the journal "Biology".
Authors:
The present version has been amended by Mrs Simin Jashmidi, in addition the page charges include English correction by the journal. We expect therefore that the final version would be significantly improved.
Reviewer:
There are some spelling mistakes, for example a space is always placed between number and unit (line 133,. But also with the hyphen (line 130) and the and equal sign (line 180). Please carefully revise the manuscript with regard to these small spelling mistakes!
Authors:
We did ou best to correct those spelling mistakes, and expect that further editing by the journal would remove any remaining.
Reviewer:
I think the supplementary figures are worth showing. If at all possible I would include part of the supplementary figures in the main manuscript.
Authors:
We think it will complicate too much the revision of the article as more “discussion” of the figures may be requested/welcome then. In addition, it rises the issue of colors, which may increase consequently the cost of publishing. Finally, we think that for part of the audience these supplementary figures would not be required for understanding the main message. These figures add arguments to the text, or provide information required for those who are not familiar with the subject. For example, following reviewer’s request a glossary has been introduced. Similarly a decision making chart is proposed.
One more modification
We include an additional Author Dr Noureddine Hammad. He participated to earliest discussions about Warburg effect, and for this reason his name was included in the earliest manuscript project. However due to negligence his name disappeared in subsequent draft at the origin of the present opinion paper. We respectfully request the acceptance of what may be seen as a late modification of authorship but is in fact the restoration of the initial state.